# The Effect of Mechanical Activation of Fly Ash on Cement-Based Materials Hydration and Hardened State Properties

**DOI:** 10.3390/ma16082959

**Published:** 2023-04-07

**Authors:** Kenzhebek Akmalaiuly, Nazerke Berdikul, Ina Pundienė, Jolanta Pranckevičienė

**Affiliations:** 1Department of Construction and Building Materials, Satbayev University, Satbayeva Str. 22, 050013 Almaty, Kazakhstan; k.akmalaiuly@satbayev.university (K.A.); n.berdikul@satbayev.university (N.B.); 2Laboratory of Concrete Technology, Institute of Building Materials, Vilnius Gediminas Technical University, Linkmenų Str. 28, LT-08217 Vilnius, Lithuania; jolanta.pranckeviciene@vilniustech.lt

**Keywords:** fly ash, mechanical activation, particle size, cement, hydration, compressive strength

## Abstract

Fly ash from coal represents the foremost waste product of fossil fuel combustion. These waste materials are most widely utilised in the cement and concrete industries, but the extent of their use is insufficient. This study investigated the physical, mineralogical, and morphological characteristics of non-treated and mechanically activated fly ash. The possibility of enhancing the hydration rate of the fresh cement paste by replacing part of the cement with non-treated and mechanically activated fly ash, and the hardened cement paste’s structure and early compressive strength performance, were evaluated. At the first stage of the study, up to 20% mass of cement was replaced by untreated and mechanically activated fly ash to understand the impact of the mechanical activation on the hydration course; rheological properties, such as spread and setting time; hydration products; mechanical properties; and microstructure of fresh and hardened cement paste. The results show that a higher amount of untreated fly ash significantly prolongs the cement hydration process, decreases hydration temperature, deteriorates the structure and decreases compressive strength. Mechanical activation caused the breakdown of large porous aggregates in fly ash, enhancing the physical properties and reactivity of fly ash particles. Due to increased fineness and pozzolanic activity by up to 15%, mechanically activated fly ash shortens the time of maximum exothermic temperature and increases this temperature by up to 16%. Due to nanosized particles and higher pozzolanic activity, mechanically activated fly ash facilitates a denser structure, improves the contact zone between the cement matrix, and increases compressive strength up to 30%.

## 1. Introduction

In the search for sustainable alternatives to construction materials, two main directions exist, which can reduce CO_2_ emissions, air pollutants and wastes: recycling aggregates in place of non-renewable ones, and the employment of supplementary cementitious materials (SCMs) such as pozzolanic industrial waste—primarily fly ash—to partially replace Portland cement [1]. Fly ash, along with other well-known pozzolanic additives, has been widely used in concrete in place of ordinary Portland cement (OPC), which has become popular today [2]. Fly ash is a byproduct of burning coal primarily collected at the top of boilers in coal-fired power plants [3]. Under the conditions of the combustion regime of boilers, most of the mineral substance of the fuel passes into ash and a minor part into slag. Almost all produced granulated blast furnace slag is used in concrete production, whereas fly ash recycling is only 30%, mainly due to its largely variable properties from using different fuels and firing technologies [4]. According to ASTM [2,5], fly ash is classified into two classes: “C” and “F”. In fly ash, the “Class F” amount of SiO_2_, Al_2_O_3_, and Fe_2_O_3_ exceeds 70%, whereas in “Class C”, the amount of SiO_2_, Al_2_O_3_, and Fe_2_O_3_ varies between 50% and 70%. ”Class F” fly ash is a typical pozzolan, mainly consisting of silicate glass supplemented with iron and aluminium [6]. “Class F” fly ash is mostly amorphous but also contains crystalline phases such as quartz, mullite, iron oxides, lime, and periclase [7]. Usually, the crystalline constituents of fly ash are non-reactive. The amount of reactive SiO_2_ must be between 40 and 50% of the total mass, and reactive Al_2_O_3_ amount is also essential [8].

Since the number of reactive oxides is rarely specified, comparing different research results is challenging. The amorphous phase’s amount and composition significantly impact “Class F” fly ash reactivity in an alkaline environment. A higher amorphous phase content causes faster activation and increased reactivity [9]. Nevertheless, this content can vary considerably due to the variability in the composition of fly ash and the ratio of amorphous to crystalline phases. It must be pointed out that the ash’s reactive phase does not entirely correspond to the amorphous phase of ash. For example, some refractory phases, such as mullite, are not reactive but are also considered to belong to the fly ash amorphous phase. Since the CaO content in “Class F” fly ash is less than 10%, the formation of calcium silicate hydrate (CSH) through the pozzolanic reaction requires a cement which, during hydration, forms portlandite. Consequently, the effectiveness of fly ash in concrete is mainly determined by its chemical components [10].

There are many benefits to using fly ash in place of Portland cement [11]. Adding fly ash to concrete improves workability [12] because the fly ash particles are spherical. Additionally, incorporating fly ash in concrete lessens bleeding and segregation [13]. The addition of fly ash in the cement paste due to the dilution effect, which reduces the flocculation of particles and the lubricating effect of the smooth surface of fly ash particles, can improve the rheological properties of fresh paste [14]. Fly ash’s pozzolanic reaction often takes longer than the hydration of cement [15,16]. The use of fly ash in the cement paste results in a pozzolanic interaction between the amorphous phase of fly ash and the Ca(OH)_2_ produced during the hydration of cement, which produces more C-S-H gel and increases density and strength [17]. Consequently, adding fly ash can cause the mass concrete’s temperature to drop [15,18].

Mechanically activated fly ash greatly increases the performance of cement-based materials compared to fly ash without mechanical activation. Additionally, cement-based materials’ pore structures become denser due to fly ash’s pozzolanic reaction, which can significantly reduce permeability [19]. According to research [20], the use of milled fly ash in paste, mortar, and concrete, due to the strengthening of the interfacial transition zone, allows the strength activity index and resistance to corrosion for concrete to significantly increase when compared to fly ash that has not been activated. Scientists [21] examined the water permeability of recycled aggregate concrete and concluded that replacing the cement in recycled aggregate concrete with milled fly ash significantly impacted the reduction in the water permeability coefficient.

Class F fly ash, due to its more stable chemical composition than Class C fly ash, high silica amount, and pozzolanic properties, is widely used for cement replacement in concrete [22,23]. As is pointed out in the literature [1,24], the size of fly ash particles varies depending on the combustion method, coal source, and other factors, ranging from less than 1µm to greater than 200 µm, but Class C fly ash is usually finer than Class F fly ashes, owing to the larger amounts of alkali sulphate present in high-Ca ashes [25]. The particle size of fly ash largely determines its activity. In addition, the mechanical activation of fly ash by milling is one of the simplest measures of increasing its surface area, thereby causing a surge in fly ash reactivity [26,27].

A milling technology called mechanical activation uses mechanical energy (such as high-energy ball milling and disintegration) to raise the reactivities of materials such as fly ash, and then to speed up the chemical reactions between solids and/or liquids [28,29]. Mechanical activation greatly facilitates fly ash decomposition [30]. Research shows that the smaller the particle size of fly ash, the higher its hydration activity and hydration rate. Therefore, to make better use of fly ash, it is usually refined by mechanical activation methods [16,31]. Research results on blended cement and concrete containing ultrafine fly ash with an average particle size of less than 10 µm or a specific surface area of more than 700 m^2^/kg exhibit improved performance over a comparable control mixture. These improvements include the filling and water-reducing role, pozzolanic reaction, and improvement roles of mechanical and durability properties [32,33]. It is important to note that mechanical activation can successfully increase the solubility of several mineral crystals [34].

Furthermore, some publications [35,36] have suggested that mechanical activation might make fly ash recycling easier and more effective because modification of the surface of fly ash particle through mechanical activation additionally decreases the crystallinity of fly ash minerals. However, the effects of mechanical activation on fly ash modification are still unknown and need to be thoroughly investigated. Promising results were presented in the research of [37], where mechanical activation was combined with chemical activation to treat hybrid fly ash cement, and it was considered a promising method to improve slow the strength development and early-age strength of concrete containing fly ash. According to researchers [11,38], mechanical activation is a promising strategy that can be used with fly ash to achieve maximal bulk utilisation with significant value addition.

A literature review shows that the mechanical activation of fly ash is a promising method to improve the ash reactivity and early strength growth of cement-based materials. However, very little attention is paid to how the mechanical activation of fly ash affects the cement minerals dissolution process, the transition of ions into solution, spreadability, setting time, and the exothermic temperature of cement paste. Only a limited number of papers have focused on the hydration of mechanically activated fly ash and cement system. This contribution aims to study the effect of mechanical activation on the fly ash particle size, pozzolanic properties, and their impact on the rheological properties and the exothermal profile of cement paste.

## 2. Materials and Methods

### 2.1. Raw Materials

Portland cement CEM I 42.5 N (PC), complying with EN 197-1: 2011 [39] requirements, was used in the present research. The mineral content of the PC by weight was C_3_S—58.54%; C_2_S—15.29%; C_3_A—10.4%; Ca_4_AF—10.17%. The specific surface was 0.41 m^2^/g. The chemical composition of the PC by weight was: SiO_2_—20.76%; Al_2_O_3_—6.12%; Fe_2_O_3_ –3.37%; CaO—63.5%; K_2_O + Na_2_O—1.03%; loss on ignition—0.30%. The specific surface (Blaine fineness) of the OPC was 4010 cm^2^/g, bulk density was 1.1 g/cm^3^, setting started at 140 min, setting ended at 190 min, and the amount of alkali was max 0.8%. The size of the OPC particles ranged from 0.1 to 70 μm. In terms of the particle diameter, 10% was 0.69 μm, 50%—8.31 μm, and 90%—31.89 μm. The mean particle diameter was 12.58 μm particles (Figure 1).

The fly ash used in the research was obtained from a coal-fired power plant. The fly ash of coal has a predominantly aluminium silicate composition. The average oxide composition (main components) of as-received untreated fly ash (UFA) is presented in Table 1. According to the ratio of the sum of Fe, Ca, Mg, Na, and K oxides to the sum of Si, Al, and Ti oxides, they are divided into acidic (less than 1) and basic (more than 1). The ash of coals is mainly acidic; of combustible shale and wood—basic. In our case, this ratio was 0.033, and it can be concluded that UFA belongs to acidic FA.

It can be seen that the fly ash used in this study can be defined as “High Calcium” fly ash or “Class C” FA, since the CaO content is higher than 25% [40].

The specific surface of the UFA was 3710 cm^2^/g, the density was 2.2 g/cm^3^, and the bulk density was 0.794 g/cm^3^. It can be seen that the specific surface of UFA is slightly lower than that of cement (PC). The loss on ignition at 1000 °C was 5.2%. According to Figure 2, where UFA is observed by scanning electron microscope (SEM), it can be concluded that most of the UFAs are spheres with an uneven porous surface and a diameter of 80–200 µm. It can also be seen that smaller spheres have a smoother surface. As is pointed out in the literature, most of the ashes have a spherical particle shape and a smooth vitrified surface texture [41]. Such a surface, which we can see according to Figure 2, indicates a high combustion temperature; the porous surface of spheres is formed by the release of gaseous compounds. Generally, the size of UFA particles does not exceed 300 μm. Particle diameter at 10% was 11.53 μm, 50%—71.11 μm, and 90%—160.82 μm. The mean particle diameter was 80.93 μm (Figure 3).

The XRD results of UFA are presented in (Figure 4). The mean phase mineral composition of UFA includes mullite (3Al_2_O_3_·2SiO_2_) and quartz (SiO_2_). As the research of [42] points out, the main phases of fly ash are quartz and mullite. Some other minerals, such as magnetite (Fe_3_O_4_), anhydrite (CaSO_4_), anorthite (CaAl_2_Si_2_O_8_), and lime (CaO), can be identified, but it depends on the geological factors related to the formation and deposition of coal. The broad hump (7–30 θ) proves that mainly SiO_2_ is presented in the amorphous phase of the UFA [27].

For obtaining the mechanically activated fly ash (AFA), the as-received fly ash was ground by a laboratory planetary ball mill for 7 h. In the milling chamber, the ratio between the ball and ash was set as 0.275. After mechanical activation, the specific surface of the AFA, compared to UFA, increased by 2740 cm^2^/g and reached 6450 cm^2^/g; the density increased by 0.33 cm^2^/g and 2.53 cm^2^/g; and the bulk density was 0.660 g/cm^3^. Reasonably close results were obtained in other studies [11,18]. The average oxide composition (main components) of AFA is presented in Table 2. The data in Table 2 show that the amount of SiO_2_ and Al_2_O_3_ in the AFA composition increased by about 1–1.5% after mechanical activation. Other researchers [43,44,45] also noticed such an effect. The mechanical activation process causes the fragmentation of coarse particles, increases the specific surface area, and increases the amount of Si and Al in the fly ash. Increasing the particle fineness and amorphous phase leads to an increased amount of silica and alumina.

The detailed particle size distributions of AFA powder are presented in Figure 5. The size of mechanically activated fly ash particles does not exceed 30.73 μm. Some very fine particles (about 40 nm) are observed after mechanical activation. Of the particle diameter, 10% was 0.92 μm, 50%—3.72 μm, and 90%—14.73 μm. The mean particle diameter was 6.06 μm. It is seen that mechanical activation decreases the particle mean diameter by almost 13 times. The morphologies of AFA after mechanical activation are shown in Figure 6a. Practically all UFA particles were crushed into sharp-edged pieces, whose dimension (Figure 6b,c) was around 1–2 μm, and only a tiny number of particles remained spherical.

The XRD results of AFA are presented in (Figure 7). The mean phase mineral composition of AFA is the same as for UFA, but some decrease in crystallinity is observed. The amorphization of crystals (the declining peak intensity due to continuous collisions and activating crystals) is a common phenomenon generated by ball milling [46]. The same observation is pointed out in the research of [47]. A slight decrease in peak intensity and broadening of quartz and mullite was detected in the sample milled for 7 h (Figure 7), compared to the XRD patterns of the UFA sample (Figure 4). This is due to the damage of crystalline structure during intense mechanical activation, which resulted in micro stress, structural changes in Si–O–Si bonding intensity, and peak broadening [35,48]. However, for various milling times, notable variations in the width and height of several peaks have been noted [42]. With an increased milling time, certain peaks lost part of their intensity, especially their touching quartz and mullite phases. The findings showed that the milling process contributed to crystal planes’ preferential orientation and increased the amorphous SiO_2_ phase, which boosted pozzolanic activity [29].

### 2.2. Preparation of Water Suspensions, Pastes, and Samples

The necessary amount of 20% UFA or AFA was selected based on a literature review observation of previous work [42], because a higher amount can cause significant retardation of cement hydration. In all cases, the tests were designed to evaluate the efficiency of the mechanical activation of fly ash. For this purpose, the effect of the UFA and AFA on the PC paste EC and pH, and the influence on the spreadability, setting time, and EXO profile (evaluated in detail cement hydration process) was conducted.

To better describe how mechanical activation affects the EC and pH values of AFA aqueous suspension, tests of UFA and AFA in distilled water at a temperature of 20 °C were performed. The effect of different amounts of UFA and AFA on the values of the EC and pH of the cement paste was investigated at 20 °C. For this purpose, pure PC (K-0) and 8 paste compositions for every amount of UFA (5–20%) and AFA (5–20%), with a stable solid–water ratio of 1:5, were prepared (Table 3).

To study the effect of adding different amounts of UFA and AFA on the process of cement hydration (setting time, spreadability, EXO profile), the same pure PC (K-0) and 8 paste compositions with different amounts of UFA and AFA, but a W/C ratio of 0.27, were prepared (Table 3). 

The weighted components were mixed in a planetary mixer. PC with UFA or AFA was mixed at a speed of 28 rpm for 5 min; then, water was poured, and the paste was mixed by forced mixing at 56 rpm for the next 5 min. The temperature of the components and the mixing environment was 20 ± 1 °C.

### 2.3. Test Methods

The UFA and AFA powder analyses were performed using a high multichannel performance sequential Wavelength Dispersive X-ray Fluorescence (WD-XRF) spectrometer (AXIOS-MAX, Panalytical, Eindhoven, Netherlands). The WD-XRF system operated for 1440 s. The quantitative analysis was performed using “Omnian” software, and corresponding standards were used for a standard-less analysis.

The X-ray diffraction (XRD) analysis patterns of the studied UFA and AFA were analysed using a diffractometer DRON-7 (Co anode and Fe filter). The measurements were conducted in the 2θ range of 5–60°. Phase identification was performed using the reference of the ASTM database.

Chapelle’s method was used to assess the pozzolanic activity of the UFA and AFA following the standard NF P18-513 [49,50]. Based on the required characteristics, the hydraulic activity of the UFA was determined per EN 197-1:2011 [39]. Based on the chemical constitution of the examined UFA and the calculation techniques described in [51], the hydraulic index K3 was derived. The ratio of basic to acidic oxides, expressed as (%CaO + %MgO + %Al_2_O_3_)/(%SiO_2_), is the hydraulic index K3. Good hydraulic properties are indicated by K3 values greater than 1.0.

The effect of the mechanical activation of UFA on the electric conductivity (EC), and the pH values of UFA and AFA suspension and PC pastes with UFA and AFA, was investigated by using the MPC 227 instrument manufactured by “Mettler-Toledo” (Columbus, OH, USA, pH electrode InLab 410, measuring accuracy 0.01; EC electrode InLab 730, measuring range 0–1000 mS/cm). The EC and pH values of the prepared FA and AFA suspensions (solid material to water ratio 1:5) and PC pastes with UFA and AFA, with a solid–water ratio of 1:5 (Table 3), were measured at 20 °C.

The morphology of the UFA and AFA was observed with a scanning electron microscope (SEM) JSM-7600F JEOL (resolution 2.0 nm).

For the mini-slump test, the paste was filled into a Vicat cone (height 40 mm, upper diameter 70 mm, bottom diameter 80 mm) and placed on a glass plate. Lifting the cone allowed the paste to spread.

Paste setting parameters were tested according to EN 196-3:2016 [52] using the Vicat test procedures.

The hydration characteristics of the fresh OPC paste were measured by an exothermic profile per ALCOA methodology [53] and evaluated according to [54]. The heat generated in exothermic reactions of OPC minerals in fresh OPC paste was measured at 20 °C, using 1.5 kg specimens placed in an insulated 10 × 10 × 10 cm textolite mould. A thermocouple (type T) was embedded in the specimen and linked to a data capture system to record the temperature as a function of time.

Compressive strength tests were performed with an ALPHA 3-3000 S testing machine. The paper presents the arithmetic averages of each of the five successful measurements. The specimens were made and cured per EN 196-1:2016 [55]. Specimens were moulded by vibration. The specimens were kept in moulds for 1 day in normal conditions and then cured in water at (20 ± 1) °C for 27 days. They were tested after 2, 7, and 28 days.

## 3. Results and Discussions

### 3.1. Pozzolanic Activity

The calculated hydraulic index K_3_ of UFA is 0.58, indicating that is has poor hydraulic properties. Existing techniques for calculating fly ash reactivity include measuring heat release and calcium hydroxide consumption in high pH environments [56], or the method based on measuring fly ash dissolving in sodium hydroxide at various temperatures (20–40 °C) [57,58].

Regarding pozzolan materials, fly ash is used as a partial cement replacement where the Ca(OH)_2_ reacts with the aluminate–silicate phase to produce calcium–silicate–aluminate hydrates [59,60]. This happens during the cement hydration associated with SiO_2_ and Al_2_O_3_ in the amorphous form. The pozzolanic activity of the UFA is 721.7 mg of CaO/g. It indicates that the UFA is rich in amorphous SiO_2_, which can react with Ca(OH)_2_ to create additional hydration products, such as C-S-H, which improves the physical and mechanical properties of the cementitious system. SiO_2_ is usually formed at high temperatures during the incineration of coal. The mechanical activation significantly increases the pozzolanic activity of the AFA—1389 mg of CaO/g. Several studies have pointed out that mechanical activation increases pozzolanic activity [2,61]. The mechanical activation process causes the break-up of coarse particles, increases the specific surface, increases the amount of Si and Al in fly ash composition and, consequently, increases the pozzolanic activity [43]. As pointed out in the study [62], pozzolanic reactivity is directly proportional to the fineness of fly ash. In our case, after mechanical treatment, the mean diameter of AFA decreased by 13 times compared to UFA. Additionally, about 2–3% of AFA particles are 40 nm; as stated in [63], this can significantly increase pozzolanic activity. The pozzolanic activity of both used UFA and AFA was confirmed by the results of the activity index of the prepared specimens (Table 4). 

It was found that used UFA possesses low activity because just a small amount (5%) of cement replacement by UFA shows a reactivity index after 2, 7, and 28 days higher than 1. This fact is very interesting because it shows that a small amount of UFA is more valuable than a higher amount of UFA. As is known, fly ash needs a highly alkaline environment created by cement dissolution to activate the pozzolanic reaction. It seems that 5% of cement replacement by UFA has little influence on the alkaline environment and improves cement mineral dissolution so that the rate of hydration and pozzolanic reaction increases [64].

It was found that AFA amounts of 5–20% used for cement replacement are of high activity, since the activity index after 28 days of sample hardening is greater than 1 (Table 4). The best results for AFA pozzolanic activity were obtained when 5 and 10% of the cement was replaced with AFA, which was about 23% and 33% higher than the pure cement sample. In the case of UFA, higher activity was determined when UFA replaced 5% of cement. A study [57] showed that the milled fly ash accelerates the cement hydration process, densifies the cement matrix, and improves the compressive strength, compared to the paste with untreated fly ash. The role of the nanosized particle of pozzolanic materials was shown in the study of [65]. The presence of nanosized particles accelerates the formation of C-S-H and C-A-S-H, which is related to increasing an initial consumption of calcium hydroxide in the pozzolanic reaction and as a sequence leading to enhancing the compressive strength of hardened cement paste. These hydrates (C-S-H and C-A-S-H) act as good binding centres between the remaining cement grains [66]. C-A-S-H spatially contributes to compressive strength growth. 

Incorporating nanosized particles of AFA (about 2–3% of AFA particles, as is pointed out in Figure 5) and replacing up to 10.0% of cement with AFA leads to activity index values increasing up to 1.33 after 28 days of hardening.

### 3.2. EC and pH Tests of Suspensions and Pastes

The EC and pH measurements of the prepared suspension (solid material to water ratio 1:5) were conducted to better understand the behaviour and dissolution of UFA and AFA in water. The results show that both UFA and AFA show a little acidic pH and a slight decrease after mechanical treatment, which is related to an increased amount of SiO_2_ in composition (Table 5). We can see that AFA had a 3.4 times higher EC than the EC of UFA. The same results were reported in the study of [67]. It is known that EC depends on the number of ions in the solution, and it is evident that mechanical activation at the same time increases the specific surface and pozzolanic activity of AFA and facilitates the transition of ions into suspension. Observed differences in UFA and AFA are reflected in the cement paste EC and pH (Figure 8a,b and Figure 9a,b).

To better assess the impact of the same amounts of UFA and AFA, measurements of EC and pH were performed within 30 min. Due to the significant and fast increase in the EC during this period, this shows the process of the dissolution of cement minerals and the penetration of ions into solution. Supplementary materials had the most significant effect on the EC values in the early hydration period because the appearance of crystal germs stabilised, and EC values were not increased. Later, the reduction in EC, due to the growth of hydrates, was observed [68]. Immediately after mixing, the highest EC had pure PC suspension (K-0), and the EC reached 15.1 mS. As can be seen, UFA influences the PC dissolution process immediately after mixing (Figure 8a). The replacement of cement by 5–20% by UFA decreases EC values to 12.2 mS, respectively, by 19.2%. This result shows a good relationship between EC values in suspension and cement replacement by UFA. The obtained EC reduction results confirms that lower amounts of cement ions are passed into suspension.

During the first 5 min, the less UFA in the suspension, the greater the increase in the EC of paste. If in the pure PC suspension, the EC value increases by 10% and the EC of UFA-20 suspension increases by 3%. After 30 min, the observed tendencies remain, and the EC value of pure PC suspension reached is 17.7 mS, whereas the EC of UFA-20 suspension reaches 14 mS. Equal amounts of AFA have different impacts on the transition of PC ions to the suspension processes. The highest EC had AFA-5 suspension (15.4 mS), and the lowest was AFA-20. However, the EC values were 6% higher than those of UFA. Compared to pure PC suspension, replacing cement with 5–20% of AFA decreases EC by 14%. This can be related to the fact that the EC value of mechanically activated AFA is significantly higher than UFA (Table 5) and contributes to the overall EC values of the suspension [69]. According to other findings [30], fly ash significantly contributes to EC growth when the paste is kept at 40 °C. The presented results show that UFA, with low EC values in water suspension, lowers the EC values of the cement suspension and retards the cement hydration process. AFA, with 3.3 times higher EC values in water suspension, retards the cement hydration process less than UFA. The cement hydration process is promoted when AFA replaces 5% of cement.

The pH values in the pure PC suspension within 30 min increased from 12.92 to 13.7 (Figure 9a). Contrary to EC tests, UFA amounts (5–20%) have minor effects on the initial pH in the PC suspension. The initial pH values reduced from 12.92 to 12.81 when the UFA amount increased to 20%. A decreased pH value occurred because less cement was used, and UFA had an acidic reaction in water suspension. As in the EC tests, it can be observed that the higher amount of UFA employed in suspension, the lower the obtained pH values after 30 min.

The impact of AFA on the suspension pH values on the mechanical activation is presented in (Figure 9b). It can be seen that an increase in AFA amount in the suspension had a slightly greater decrease in pH values compared to the suspensions with UFA. The initial pH values reduced to 12.79 when the AFA amount increased to 20%.

The results of the mini-slump tests were obtained immediately after the cement paste mixing (Figure 10). The impact of the UFA amount on the spreadability of fresh cement pastes is manifested by a significant increase in the measured paste spread diameters. As is pointed out in [31,70], the more spherical the particle content of fly ash particles, the less water that is required and significantly influences the workability [31]. When the UFA amount increased to 20%, the paste spread diameters increased by 23%, reaching 13.5 mm. This is due to the spherical shape of UFA particles and the fact that less cement was used; additionally, UFAs particles are 6.5 times larger than cement particles [71]. The spherical shape of UFA particles can improve the rheological characteristics of freshly made pastes and reduce the surface friction between cement particles [72]. The mechanical activation of AFA reduces the particle size, so they are smaller than cement particles at least by two times, and the spreadability of fresh cement pastes decreases. When the AFA amount increased to 20%, the paste spread diameters decreased by 18% and reach 9.0 mm.

The conclusion is that at low amounts (5%) of UFA and AFA, the spreadability of fresh cement pastes changes a little. Higher amounts of UFA increase spreadability, but higher amounts of AFA, on the contrary, decrease.

### 3.3. Setting Time

The results of the initial and final setting times of fresh cement pastes are presented in Figure 11 and Figure 12. The initial setting time results show that with the increase in cement replacement by UFA amount until 20% in the paste, the initial setting time increased by three times—from 80 to 240 min. This result corresponds to the EC test results (Figure 8a,b and Figure 9a,b) and the spreadability test of fresh cement pastes (Figure 10). Decreasing EC and pH values in pastes with increasing UFA or AFA reflect increasing spreadability and prolonged initial setting time. Lower amounts of UFA (5%) do not significantly influence the initial setting time, but replacing 10–20% of cement with UFA increases the initial setting time from 2 to 3 times. The same tendencies were observed in the research [73]. In the case of AFA, the initial setting times of fresh cement pastes showed the same trends noticed in research [11]—with an increased amount of cement replacement by AFA, the initial setting time increased. However, when cement replacement by AFA was 5–10%, the initial setting time was 4.5–6.7% lower than in the same paste with UFA, but when cement replacement by AFA increased until 20%, the initial setting time was shorter by 10–17% than in the same composition of pastes with UFA. The increased AFA amount shortens the setting time compared to UFA’s initial setting time. This means that more water is distributed among the AFA particles and, correspondingly, less water interacts with the cement particles; in this way, hydration is prevented. As is pointed out in the literature [14,74], the rheological properties of pastes with different fineness of fly ash are highly related to the fly ash particle size. The mechanically activated fly ash can play a vital role in reducing the water requirement of normal consistency and decrease in setting time, due to the fineness of particles reducing the porosity in paste.

The final setting time measurement results showed that an increased amount of UFA until 20% in fresh cement pastes gradually increased the final setting time (prolonging it from 120 to 270 min). The final setting time did not significantly change when the AFA amount was 5–10%. When 20% of AFA was used in the pastes, the final setting time increased to 215 min. Compared to paste with UFA, the highest amount of AFA (20%) significantly decreased (by 20%) the final setting time. Presumably, this reduction is influenced by increased EC in pastes with AFA.

We can conclude that UFA and AFA retard the hydration of cement minerals because their presence decreases the EC and pH of fresh cement pastes, not only due to the lower amount of cement, but also because fly ash prevents cement minerals dissolution in the early stage of hydration. The lower pH values of UFA and AFA also affect and decrease the pH of fresh cement paste. All measurements of pozzolanic activity, pH and EC, spread, and setting time of fresh cement pastes correlated. The results of pozzolanic activity, spread, and setting time of fresh cement pastes show the same tendencies as in the study [69].

### 3.4. EXO Profile

To determine whether the heat emitted during cement hydration affects the temperatures of pastes, fresh cement pastes with various amounts of UFA and AFA were examined (Figure 13a–c and Figure 14a–c). The change in exothermic reaction maximum (EXO) temperature and EXO time in pastes depending on the UFA and AFA amounts are reflected in research on thermal profiles during the early hydration process in fresh cement pastes (Figure 13, Figure 14 and Table 6). The test results showed that with the replacement of UFA (Figure 13a–c) until 20% in the paste, the EXO maximum time reached was prolonged from 7.8 to 10.52 h (Figure 13a). Lower pH of UFA leads to significant retardation of cement minerals dissolution, which was reflected in increasing EXO time. The EXO temperature of pure cement paste reached 73.36 °C, but when 5% of cement was replaced by UFA, the EXO maximum temperature increased rapidly until 82.95 °C. For pastes with 10, 15, and 20% cement replacement with UFA, EXO maximum temperature decreased until 61.33 to 61.09 and 59.53 °C (Figure 13c). These results show that small amounts of UFA increased the EXO maximum temperature by 12.3%, compared to pure cement past EXO maximum temperature. Small amounts of UFA seem to react more actively with cement minerals in the paste. The same tendencies were observed in the study of [11], where the effect of untreated and mechanically activated fly ash on hydration heat was investigated. The same was observed: compared with pure cement, the EXO maximum time for untreated FA and mechanically activated FA was prolonged. This might be because fly ash has acidic pH and a high specific surface area. Higher amounts of UFA (10–20%), possibly due to the lower amount of cement minerals which can react with pozzolana, decreased the temperature by 19.7–21.7%.

The test results showed that with the replacement of AFA (Figure 14a–c) until 20% in the paste, the EXO time was prolonged from 7.8 to 9.68 h (Figure 14b). Compared to the UFA results, mechanical activation somewhat stabilizes the EXO maximum time. In the paste with 5% cement replacement to AFA, the EXO maximum temperature significantly increased by 3 °C compared to the same paste with UFA. Compared to the K-0 paste results, 5% cement replacement to AFA increased the EXO maximum temperature by 15.1%.

As the study of [11] pointed out, when 20% of cement was replaced by mechanically activated fly ash, hydration heat increased by 9% compared to the results of pure cement paste. The possible explanation might be that fly ash introduced some additional readily soluble compounds, which caused the increase in the cumulative hydration heat at a very early stage. Mechanically activated fly ash has a more exposed surface area, helping to dissolve these compounds more efficiently. In our case, AFA was much finer than UFA and provided more nucleation sites for hydration. This might be caused by the pozzolanic reaction of mechanically activated fly ash. Fly ash needs a high alkaline environment created by cement hydration to activate its pozzolanic reaction. Therefore, the pozzolanic reaction of fly ash with or without mechanical activation can be assumed negligible in the very early stage of cement hydration [75]. However, mechanically activated fly ash is supposed to provide an extra surface for the heterogeneous nucleation of C-S-H gel, further promoting hydration kinetics. This phenomenon is usually called the filler effect [64]. A more noticeable increase in EXO maximum temperature was seen when AFA replaced 10, 15, and 20% of cement. The EXO maximum temperature reached 68.02, 67.28, and 66.08 °C (Figure 14c). Compared to pastes with the UFA results, the increment in temperature was 11.4, 11.8, and 10%. It is evident that mechanical activation significantly accelerates the hydration of cement minerals. Lower pH leads to significant retardation of cement minerals dissolution, which is reflected in cement paste EC values, setting time, EXO temperature and time (especially in the case of UFA). Possibly due to the more expressed pozzolanic activity of the AFA (higher specific surface, EC values in suspension), AFA more actively reacts with cement hydration products. In addition, replacing 10% of AFA avoids a significant increase in the hydration temperature.

Based on the proposed method [54], it is possible to evaluate the retarding effect of the UFA and AFA in the paste. This calculation makes it possible to evaluate the retarding effect on various parameters—varying in the temperature of the EXO effect (ET), varying in the time of reaching the EXO effect (H), and varying in the rate of temperature rise (C).

Based on the results in Table 6, the replacement of 10% of cement by UFA is marginal for cement paste compositions because higher amounts of UFA significantly deteriorate the cement paste hydration parameters. In the case of AFA, the marginal amount of cement replacement by AFA is 15%, and a higher amount is not recommended.

For a better explanation, the calculation of CaO/Al_2_O_3_, SiO_2_/Al_2_O_3_, and CaO/SiO_2_ in the compositions with UFA and AFA is presented in (Table 7). The calculation shows that the most significant decrease in the CaO/Al_2_O_3_ and SiO_2_/Al_2_O_3_ ratios occurred when UFA or AFA replaced just 5% of cement. It means that on this replacement level, Al_2_O_3_ is mostly increased, compared to CaO and SiO_2_ amounts. Possibly, this affects the increased dissolution rate of cement minerals and the increase in EXO maximum temperature. This effect is less pronounced with a further increase in cement replacement by AFA or UFA.

In general, UFA or AFA in fresh cement paste reduced the EC of the paste (to 21% when UFA was used and to 17% when AFA was used), impeded the penetration of ions to the solution, and decreased pH (to 1.6% when UFA was used and to 1.98% when AFA was used) and spread when AFA was used. Lower pH of UFA and AFA leads to significant retardation of cement minerals dissolution, which is reflected in cement paste setting, EXO temperature and time when higher than 5% of UFA and higher than 10% AFA is used. It can be stated that 5% of UFA and 15% of AFA is marginal for cement paste because higher amounts significantly deteriorate the cement paste hydration parameters.

### 3.5. Mechanical Properties

This study revealed that the mechanical activation of fly ash and different replacements of cement amounts by UFA and AFA significantly influence the compressive strength of hardened paste after 2, 7, and 28 days of curing (Figure 15). The mechanical activation of AFA resulted in a significant increase in the early strength of the sample because of nanosized particles, and an increase in Al_2_O_3_ and SiO_2_ amount after mechanical activation was observed. The highest compressive strength was obtained for the samples in which UFA replaced 5–10% of cement after 2 days of curing, but a higher amount led to a decrease in compressive strength. Compared to the pure cement paste samples, the strength increased by 7% for paste UFA-5 and 9.8% for paste UFA-10. Further increasing the replacement of cement by UFA by 15 and 20% gradually decreased the sample’s compressive strength to 19.5% and 30%, compared to the pure cement paste sample. When the cement was replaced by AFA, the highest compressive strength was obtained for the samples with 5–10% of AFA. Compared to the pure cement paste samples, the strength increased by 18% for paste AFA-5 and 22.6% for paste AFA-10.

Further increasing the replacement of cement by UFA (by 15 and 20%) showed a lower increment in compressive strength: for AFA-15 by 12.8% and by 7% for AFA-20. Higher than 10% of UFA replacement in the paste resulted in a decrease in the sample’s compressive strength; none of the used AFA amounts decreased the compressive strength of the sample, compared to the pure cement paste sample. The highest compressive strength of the sample was observed when 5, 10, and 15% of AFA were used. In the case of UFA, just 5% was used.

After 7 days of curing, observed trends remained, and on average, the strength of the specimen increased from 5 to 25%, compared to the sample strength of pure cement paste. It was noticed that increasing the amount of AFA increased the early strength after 2 and 7 days of curing more intensively than when the same amount of UFA was used. The same trend was observed in [76]. First, such an increase in strength is determined by forming a denser structure and more hydration products due to the extremely active effect of AFA. As pointed out in [77], the mechanical activation of AFA improves particle fineness, increases silica and alumina content in the treated AFA, and increases the sample’s compressive strength. When 5–10% milled FA was used [78], compressive strength was increased by 8–12%, compared to the results of untreated FA. As pointed out in [47], mechanical activation increases the reactivity of fly ash. The reactivity of fly ash varies with median particle size and increases rapidly when the particle size is reduced to less than 7 μm, resulting in the faster creation of a hydration product. Additional information about fly ash milling is presented in the research of [44,45]. The fly ash was milled together with cement because the addition of the fly ash can decrease the agglomeration effect in the milling device. During milling, the particles’ fineness was increased. Additionally, we found that the silica and alumina amount was increased. Due to increased fly ash reactivity from their mechanical activation, hardened cement paste with milled fly ash exhibited better compressive strength than cement paste with unmilled fly ash.

After 28 days of curing, the sample with 5% cement replacement by UFA increased the strength of samples by 5.8%, and a further increase in cement replacement by UFA decreased the strength of the samples by 3.7, 15.0, and 18.9%. As is pointed out in the study of [38], it was observed that by the replacement of cement with 40% of unprocessed flay ash, the compressive strength of samples after 7 days of curing decreased from 40 to 15 MPa, and after 28 days of curing from 61 to 32 MPa. Another study [29] presented results of the replacement of cement with 40% of unprocessed fly ash. If in the pure hardened paste, the compressive strength increased from 35 to 65 MPa during 2 to 28 days. When 20% of cement was replaced with UFA, the compressive strength increased from 10 to 30 MPa. When 30% was replaced, the compressive strength increased from 9 to 29 MPa; when 40% was replaced, the compressive strength increased from 7 to 23 MPa.

In the samples with AFA after 28 days of curing, an increase in the strength of the samples occurred in the following order: with increased cement replacement by AFA, the strength increased by 20.0, 25.5, 14.3, and 3.6%, compared to pure cement paste sample strength. The results show that AFA particles, due to higher pozzolanic activity and a finer particle size (less than 40 nm), promote a more significant gain in strength after only 28 days. However, the highest increase in compressive strength was obtained in the sample with cement replacement by 10% AFA. This proves there is a balance between the amount of AFA replacement and the improvement of properties, and this is associated with the possibility of a CaOH_2_ reaction in the system. In addition, it can be assumed that some stresses are taken over by the largest spherical AFA particles in this structure, surrounded by compounds of hydrated products and cement particles.

The mechanical activation of AFA results in a significant increase in the early strength of cement paste compared to paste with UFA. However, the exact mechanisms of the strength development in paste containing mechanically activated fly ash are not completely understood [79].

The incorporation of nanosized particles of AFA (approx. 2–3% of AFA particles) and replacement of up to 10.0% of cement with AFA (particles size less than 40 nm) was found [65] to accelerate the formation of C-S-H and C-A-S-H associated with an initial higher consumption of Ca(OH)_2_ in the pozzolanic reaction, thus leading to higher water resistance and an increase in cement-based material compressive strength of about 30%. These hydrates (C-S-H and C-A-S-H) act as good binding centres between the remaining cement grains. Moreover, research findings [80] showed that nano silica accelerates cement hydration, thus densifying the cement matrix and improving the compressive strength.

Thus, when the compressive strength in the pastes, with the incorporation of additional strength-providing products, increases beyond the value expected for their strength considering the dilution effect (i.e., the cement replacement level), it indicates that their pozzolanic activity prevails, promoting the improvement of mechanical properties due to the formation of additional strength-providing products, namely, C-S-H and C-A-S-H [81].

## 4. Conclusions

This study conducted electric conductivity, pH, spread, setting time, and hydration kinetics analyses of cement paste incorporating untreated and mechanically activated fly ash. The following conclusions can be made:

It was established that untreated and mechanically activated fly ash differs by pozzolanic activity more than two times, by EC values in water suspension (by 2 times), and in cement paste by 5–10%, and have a different effect on cement paste spread, setting time, and hydration course.

Pure cement paste, characterised by the highest EC values and smaller particles, compared to untreated FA, demonstrates the lowest spread. With an increased replacement of cement by untreated FA from 5 to 20% in the pastes (characterised by coarse spherical shape and 5–23% lower EC in cement paste), an increase in cement paste spread by 4.5–32% was observed.

Mechanically activated FA, characterised by a higher EC value and 13 times smaller size than UFA particles, decreases cement paste spread up to 16 % compared to pure cement paste spread.

Untreated FA, characterised by lower EC values, significantly retards the dissolution of cement minerals, which is reflected in more than twice-longer initial and final setting times of the cement paste. Mechanically activated fly ash, characterised by higher EC values and significantly smaller particle size, retards the initial and final setting times to a lesser degree. The initial and final setting times of the cement paste with untreated FA are 18% longer than in the same composition pastes with mechanically activated FA.

EXO temperature tests revealed that the same amount of untreated FA (5–20%) and mechanically activated FA (5–20%) prolongs the EXO time in the paste from 7.7 to 9.4 h. Compared to the pure cement paste sample, in the sample with 5% of untreated FA and mechanically activated FA, the EXO maximum temperature is higher by 12.3 and 15.1%. When AFA replaced 10, 15, and 20% of cement, the EXO maximum temperature was 11.4, 11.8, and 10% higher than in the same composition sample with untreated FA. It is evident that the mechanical activation of FA significantly accelerates the hydration of cement minerals, compared to untreated FA. Higher EC values, pozzolanic activity, and the smaller particle size of AFA than in untreated FA lead to a significant increase in hydration temperature, which is reflected in cement paste setting, EXO temperature, and time.

The replacement of cement by unmilled FA and mechanically activated FA had a relative influence on the mechanical strength of the hardened cement paste. The replacement of cement by mechanically activated FA resulted in 20–30% higher compressive strength at 7, 14, and 28 days than the replacement of cement by untreated FA, due to an increase of thirteen times in the fineness of the mechanically activated FA, the presence of nanosized particles, and an increase in the amount of SiO_2_ and Al_2_O_3_ in the FA composition after mechanical activation. The optimum cement replacement by untreated FA in cement composition was 5%, and for mechanically activated FA this was 10 and 15%.

## Figures and Tables

**Figure 1 materials-16-02959-f001:**
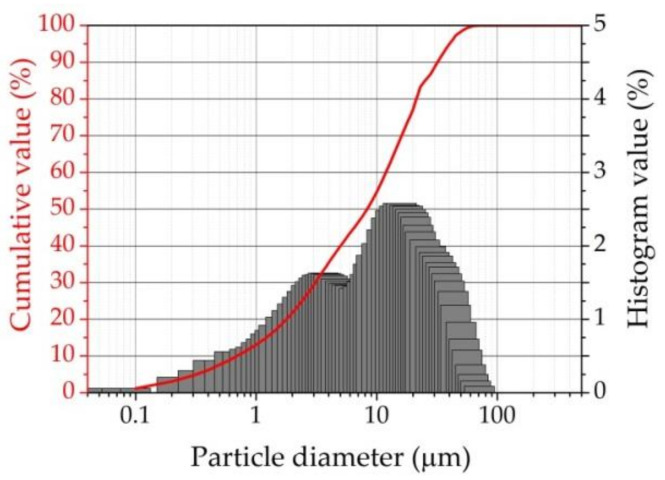
Particle size distribution of cement.

**Figure 2 materials-16-02959-f002:**
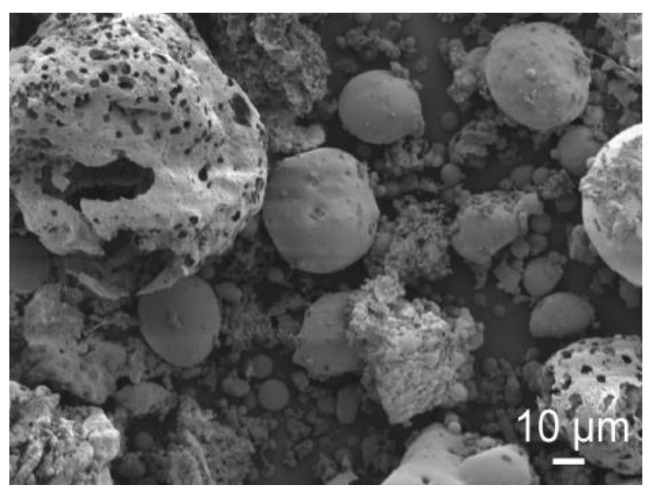
SEM image of untreated UFA.

**Figure 3 materials-16-02959-f003:**
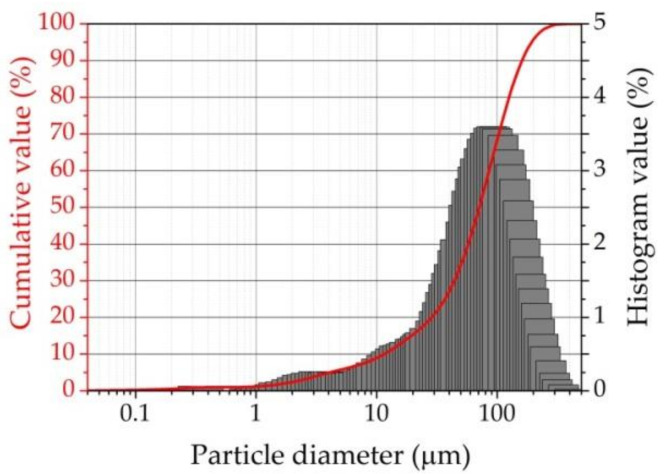
Particle size distribution of UFA.

**Figure 4 materials-16-02959-f004:**
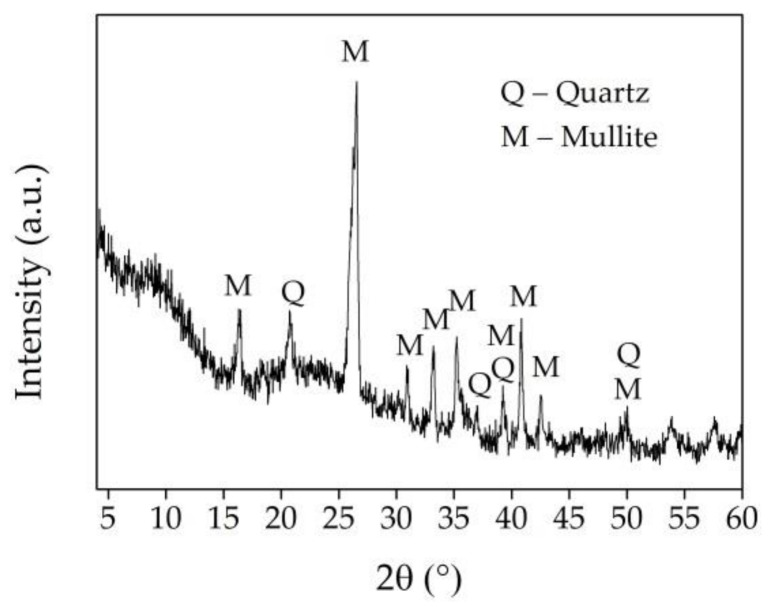
XRD curve of UFA.

**Figure 5 materials-16-02959-f005:**
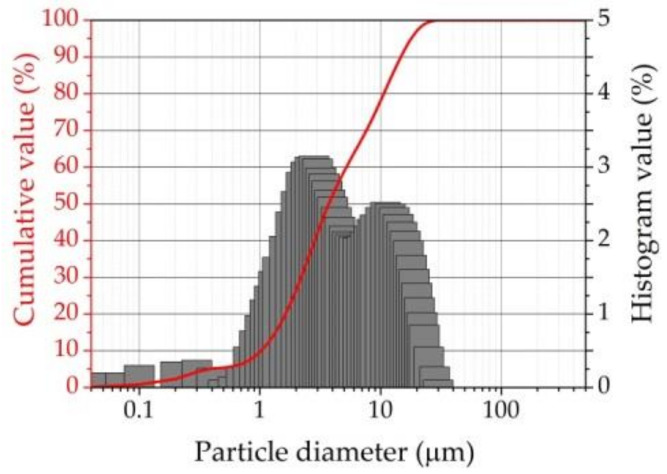
Particle size distribution of mechanically activated AFA.

**Figure 6 materials-16-02959-f006:**
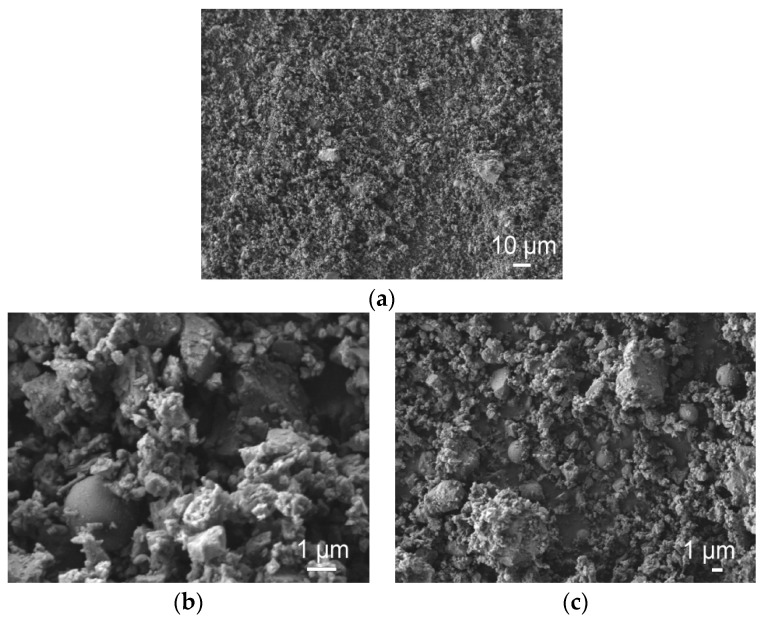
Image of mechanically activated AFA (**a**–**c**).

**Figure 7 materials-16-02959-f007:**
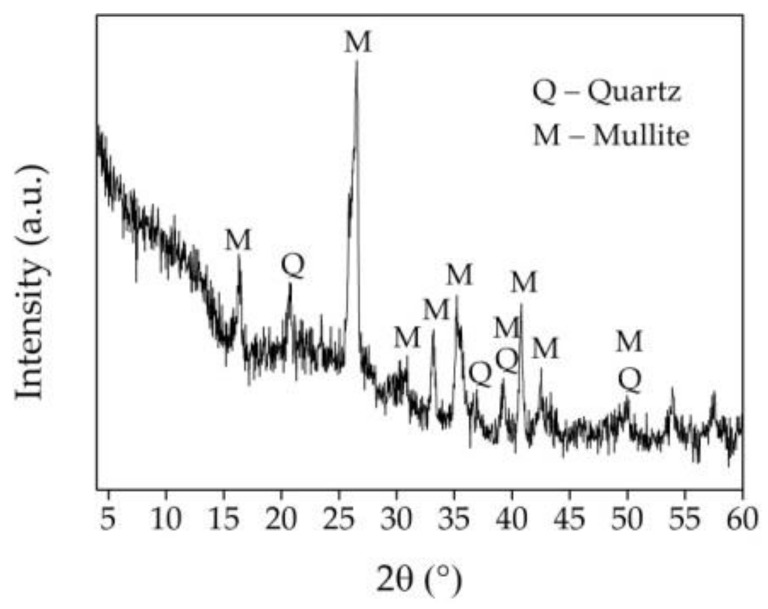
XRD of AFA.

**Figure 8 materials-16-02959-f008:**
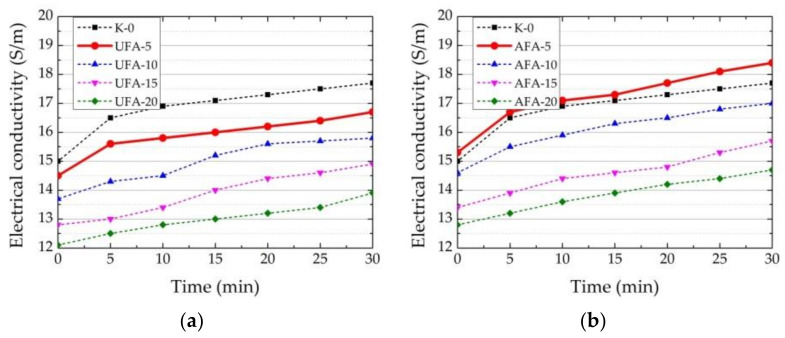
Changes in EC depending on the replacement of cement by different amounts of (**a**) UFA and (**b**) AFA in the suspension.

**Figure 9 materials-16-02959-f009:**
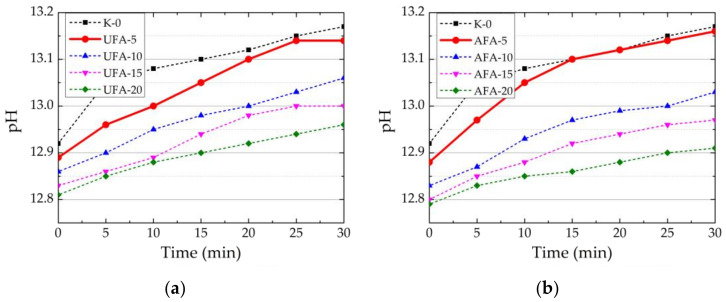
Changes in pH depending on the replacement of cement by different amounts of (**a**) UFA and (**b**) AFA in the suspension.

**Figure 10 materials-16-02959-f010:**
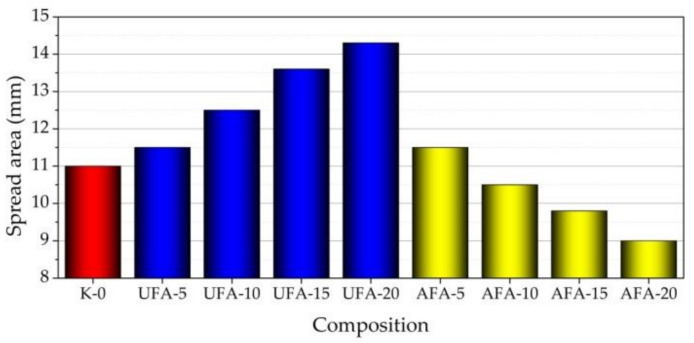
Spread area of fresh cement paste with different cement replacements by UFA and AFA amounts in composition.

**Figure 11 materials-16-02959-f011:**
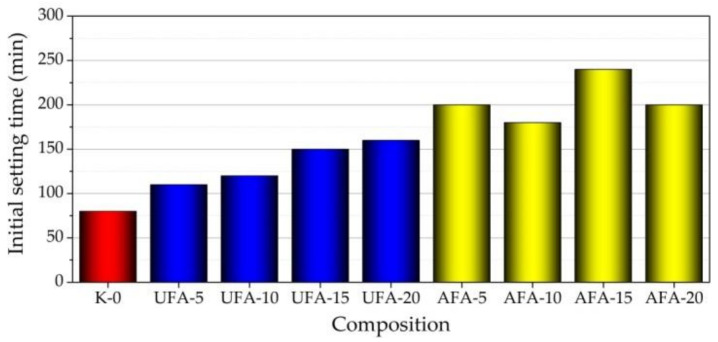
Initial setting times of fresh cement paste with different cement replacements by UFA and AFA amounts in composition.

**Figure 12 materials-16-02959-f012:**
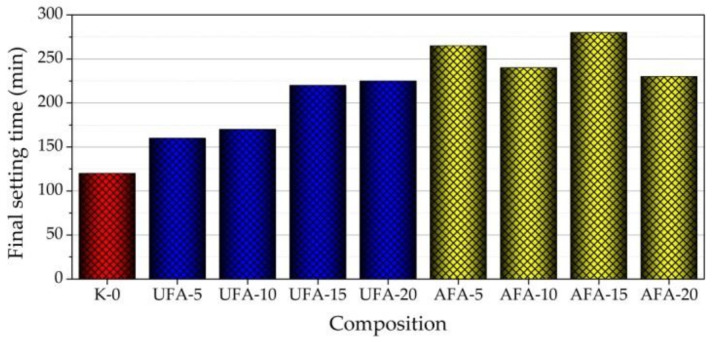
Final setting times of fresh cement paste with different cement replacements by UFA and AFA amounts in composition.

**Figure 13 materials-16-02959-f013:**
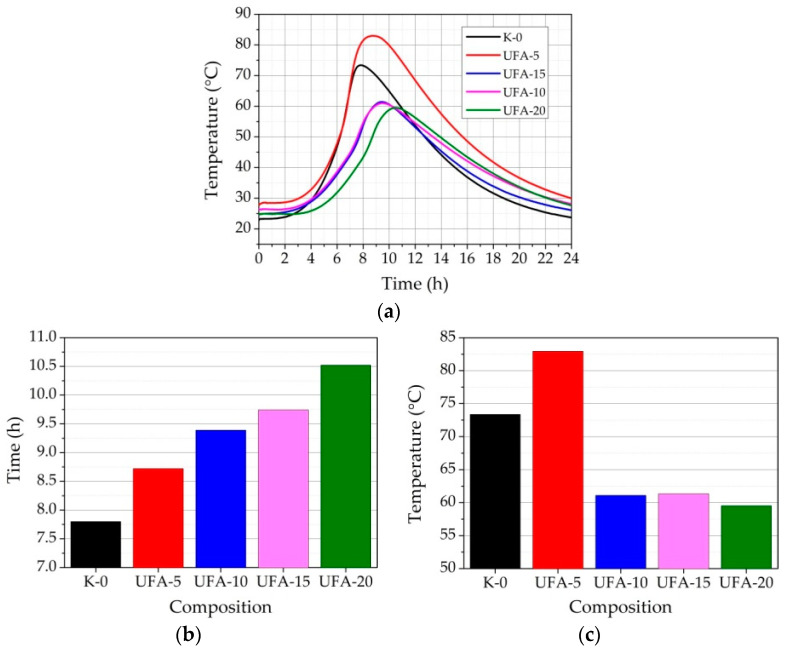
EXO profile of the fresh cement paste with different replacements of cement (**a**) by UFA in composition; the dependence of EXO maximum (**b**) time and (**c**) maximum temperature in the fresh cement paste on the amount of replacement of cement by UFA.

**Figure 14 materials-16-02959-f014:**
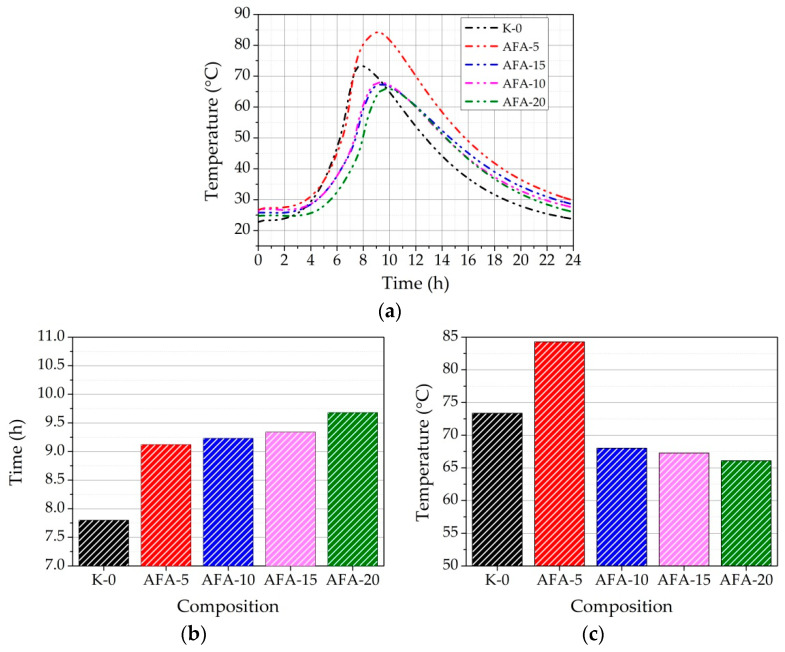
EXO profile of the fresh cement paste with different replacements of cement (**a**) by AFA in composition; the dependence of EXO maximum (**b**) time and (**c**) maximum temperature in the fresh cement paste on the amount of replacement of cement by AFA.

**Figure 15 materials-16-02959-f015:**
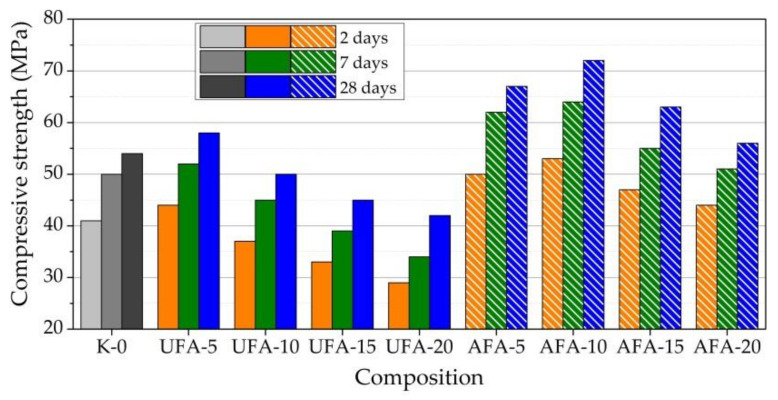
Results of compressive strength of the hardened cement pastes with UFA and AFA.

**Table 1 materials-16-02959-t001:** Chemical composition (mass%) of UFA.

SiO_2_	Al_2_O_3_	Fe_2_O_3_	CaO	MgO	Na_2_O	K_2_O	P_2_O_5_	TiO_2_	MnO	SO_3_	Loss on Ignition
57.03	31.71	1.66	1.12	0.45	0.33	1.10	0.10	1.16	0.02	0.12	5.20

**Table 2 materials-16-02959-t002:** Chemical composition (mass%) of AFA.

SiO_2_	Al_2_O_3_	Fe_2_O_3_	CaO	MgO	Na_2_O	K_2_O	P_2_O_5_	TiO_2_	MnO	SO_3_	Loss on Ignition
58.71	32.57	1.66	1.39	0.35	0.30	0.98	0.05	1.10	0.01	0.10	4.20

**Table 3 materials-16-02959-t003:** Cement paste compositions with different amounts of UFA and AFA for conductometry and setting time, spreadability, and EXO profile tests.

Composition	PC, %	UFA, %	AFA, %	S/W for Conductometry Test	W/PC for Another Test
K-0	100	0	–	1:5	0.27
UFA-5	95	5	–	1:5	0.27
UFA-10	90	10	–	1:5	0.27
UFA-15	85	15	–	1:5	0.27
UFA-20	80	20	–	1:5	0.27
AFA-5	95	–	5	1:5	0.27
AFA-10	90	–	10	1:5	0.27
AFA-15	85	–	15	1:5	0.27
AFA-20	80	–	20	1:5	0.27

**Table 4 materials-16-02959-t004:** UFA and AFA activity index.

Composition	Activity Index Values (in%) After		
	2 Days	7 Days	28 Days
UFA-5	1.07	1.06	1.09
UFA-10	0.90	0.90	0.92
UFA-15	0.80	0.80	0.85
UFA-20	0.70	0.70	0.74
AFA-5	1.20	1.24	1.22
AFA-10	1.27	1.32	1.33
AFA-15	1.14	1.12	1.15
AFA-20	1.02	1.02	1.04

**Table 5 materials-16-02959-t005:** pH and electric conduction characteristics of FA suspension.

	pH	EC, S/m
UFA	5.2	69.20
AFA	5.0	234.6

**Table 6 materials-16-02959-t006:** Parameters for assessing the retarding action of the UF and AFA.

Composition	Change in ET,%	Change in H,%	Change in C,%
K-0	–	–	–
UFA-5	+12.3	+18.1	+8.20
UFA-10	−16.4	+18.1	−30.1
UFA-15	−16.6	+20.2	−31.5
UFA-20	−17.8	+24.5	−32.2
AFA-5	+15.1	+7.30	+9.60
AFA-10	−6.80	+18.1	−23.3
AFA-15	−7.00	+18.1	−24.7
AFA-20	−9.60	+18.1	−27.4

**Table 7 materials-16-02959-t007:** Basic oxide ratio in the compositions with UFA and AFA.

Composition	CaO/Al_2_O_3_	SiO_2_/Al_2_O_3_	CaO/SiO_2_
K-0	12.7	3.91	3.27
UFA-5	9.51	3.36	2.83
UFA-10	7.42	3.02	2.46
UFA-15	5.96	2.78	2.15
UFA-20	4.88	2.60	1.88
AFA-5	9.44	3.35	2.82
AFA-10	7.34	3.00	2.44
AFA-15	5.88	2.76	2.13
AFA-20	4.81	2.59	1.86

## Data Availability

Not applicable.

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
