# Peer review of "The Effect of Mechanical Activation of Fly Ash on Cement-Based Materials Hydration and Hardened State Properties"

_materials, 2023, doi:10.3390/ma16082959_

Round 1
Reviewer 1 Report
This study investigates pozzolanic activity, EC and pH tests of suspensions and pastes, setting time, EXO profile, and mechanical properties of the composite with cement and non-treated and mechanically activated fly ash. It is validated that using the non-treated and mechanically activated fly ash can boost the hydration rate of the fresh cement. This is an interesting topic.
However, the study focuses on presenting the experimental results in section 3. I suggest the author using a separate section to discuss the role of mechanical activation of fly ash in the hydration process and in the hardened cement in combination of physical, mineralogical and morphological characteristics.
Author Response
We appreciate the reviewer’s efforts and time to help us improve the paper; answers to the reviewer’s comments and corrections in the manuscript are below.
However, the study focuses on presenting the experimental results in section 3. I suggest the author using a separate section to discuss the role of mechanical activation of fly ash in the hydration process and in the hardened cement in combination of physical, mineralogical and morphological characteristics.
Response: Thank you for the suggestion, but to present the article this way, the entire structure will have to be redone. Your valuable comment will be considered in further research articles on this topic.
Reviewer 2 Report
This article examined the physical, mineralogical and morphological characteristics of non-treated and mechanically activated fly ash. The rheological properties including spread, setting time, hydration products, mechanical properties and microstructure of fresh and hardened cement paste were examined. The article has some minor issues that needs to be addressed before acceptance.
1. Introduction section: the literatures are clustered. Adding few individual literatures regarding the rheological and mechanical properties are recommended. Please reduce the number citations used in the introduction section. More than 100 citations in this article is too much and not necessary.
2. Line 349” The replacement of cement by 349 5–20% by UFA decreases EC values to 12.2 mS, respectively, by 19.2%. Highlight the reason for this behavior.
3. Why was the percentage of UFA and AFA is restricted to 20?
4. Figure 10, Why was the spread area is higher in UFA-20 and lower in AFA-20 mixture?. Explain the reason for this behaviour.
5. Figure 11 and 12, Why was the initial and final setting time is higher in AFA mixtures compare with UFA mixtures?. Explain the reason for this behaviour. Compare the results with the literature.
6. Conclusions are clear and pertinent.
Author Response
We appreciate the reviewer’s efforts and time to help us improve the paper; answers to the reviewer’s comments and corrections in the manuscript are below.
- Introduction section: the literatures are clustered. Adding few individual literatures regarding the rheological and mechanical properties are recommended. Please reduce the number citations used in the introduction section. More than 100 citations in this article is too much and not necessary.
Response: Thanks for your comments. Several articles regarding the rheological and mechanical properties were added to the literature list.
- Line 349” The replacement of cement by 349 5–20% by UFA decreases EC values to 12.2 mS, respectively, by 19.2%. Highlight the reason for this behavior.
Response: Thanks for your comments. There is the effect of the replacement of cement by UFA. As the amount of UFA in the composition increases, the cement amount in the suspension decreases. The lowest EC values in the suspension with 20% of UFA can be explained by the lower amount of cement, that’s why respectively lower amounts of ions are passed into suspension.
- Why was the percentage of UFA and AFA is restricted to 20?
Response: Thank you for your question. The reviewed literature recommends using up to 20% fly ash because a higher amount can cause significant cement hydration retardation.
- Figure 10, Why was the spread area is higher in UFA-20 and lower in AFA-20 mixture?. Explain the reason for this behaviour.
Response: Thank you for your question. The difference in the spread area between UFA and AFA can be explained by the fact that the specific surface area of UFA particles is 2.3 times lower than the specific surface area of AFA particles. Additionally, UFA particles are spherical, and AFA is sharp-edged particles. These mentioned factors reduce the spread area of the paste with AFA. We add the sentence “The spherical shape of UFA particles can improve the rheological characteristics of freshly made pastes and reduce the surface friction between cement particles“ with reference to the article [Sun, Y.; Lee, H. Research on Properties Evolution of Ultrafine Fly Ash and Cement Composite. Constr. Build. Mater. 2020, 261, 119935, doi:10.1016/J.CONBUILDMAT.2020.119935].
- Figure 11 and 12, Why was the initial and final setting time is higher in AFA mixtures compare with UFA mixtures?. Explain the reason for this behaviour. Compare the results with the literature.
Response: Thank you for your question. Our studies show that cement paste with mechanically activated fly ash show lower spread area than cement pastes with untreated fly ash. It means more water is distributed among the AFA particles, and correspondingly less water interacts with the cement particles, preventing hydration. As pointed out in the research [Krishnaraj, L.; Ravichandran, P.T. Investigation on Grinding Impact of Fly Ash Particles and Its Characterization Analysis in Cement Mortar Composites. Ain Shams Eng. J. 2019, 10, 267–274, doi:10.1016/J.ASEJ.2019.02.001.; Snelson, D.; Wild, S.; Farrell, M.O. Setting Times of Portland Cement–Metakaolin–Fly Ash Blends. J. Civ. Eng. Manag. 2011, 17, 55–62, doi:10.3846/13923730.2011.554171.; El Fami, N.; Ez-zaki, H.; Sassi, O.; Boukhari, A.; Diouri, A. Rheology, Calorimetry and Electrical Conductivity Related-Properties for Monitoring the Dissolution and Precipitation Process of Cement-Fly Ash Mixtures. Powder Technol. 2022, 411, 117937, doi:10.1016/J.POWTEC.2022.117937] the rheological properties of pastes with the different fineness of fly ash are related to the fly ash particle size. The milled fly ash can play a vital role in reducing the water requirement of normal consistency and decrease in setting time due to the fineness of particles reducing the porosity in the paste.
- Conclusions are clear and pertinent.
Response: Thanks for your comments.